# Factors Associated with Access of Marital Migrants and Migrant Workers to Healthcare in Taiwan: A Questionnaire Survey with Quantitative Analysis

**DOI:** 10.3390/ijerph16162830

**Published:** 2019-08-08

**Authors:** Feng-Yuan Chu, Hsiao-Ting Chang, Chung-Liang Shih, Cherng-Jye Jeng, Tzeng-Ji Chen, Wui-Chiang Lee

**Affiliations:** 1Department of Family Medicine, Taipei Veterans General Hospital, Yuanshan & Su-Ao Branch, No. 386, Rongguang Rd., Yuanshan Township, Yilan County 264, Taiwan; 2Department of Family Medicine, Taipei Veterans General Hospital, No. 201, Sec. 2, Shi-Pai Rd., Beitou Dist., Taipei 112, Taiwan; 3School of Medicine, National Yang-Ming University, No. 155, Sec. 2, Linong St., Beitou Dist., Taipei 112, Taiwan; 4Department of Medical Affairs, Ministry of Health and Welfare, No. 488, Sec. 6, Zhongxiao E. Rd., Nangang Dist., Taipei 115, Taiwan; 5Department of Obstetrics and Gynecology, Kaohsiung Medical University Hospital, No. 100, Ziyou 1st Rd., Sanmin Dist., Kaohsiung 807, Taiwan; 6School of Medicine, Kaohsiung Medical University, No. 100, Shiquan 1st Rd., Sanmin Dist., Kaohsiung City 807, Taiwan; 7Department of Medical Affairs and Planning, Taipei Veterans General Hospital, No. 201, Sec. 2, Shi-Pai Rd., Beitou Dist., Taipei 112, Taiwan

**Keywords:** access, interpreter, migrant and health, marital migrant, migrant workers

## Abstract

In Taiwan, migrants come mostly for marriage and work. Several researchers have conducted health-related studies of marital migrants and migrant workers, but the access of the two groups to healthcare has not been studied. Therefore, our study investigated the factors associated with migrants’ access to healthcare, with the main foci being marital migrants and migrant workers in Taiwan. A structured and cross-sectional questionnaire was anonymously self-administered by migrants recruited to participate in this survey on a voluntary basis from 11 medical centers and 11 migrant-helping associations in Taiwan between May 1st and September 21st, 2018. A total of 753 questionnaires were analyzed. The majority of marital migrants (*n* = 243) and migrant workers (*n* = 449) surveyed were enrolled in Taiwan’s National Health Insurance system (92.7 vs. 93.5%, *p* = 0.68). More of the migrant workers (*n* = 205) than the marital migrants (*n* = 42) encountered language barriers while seeking medical services (48.0 vs. 17.1%, *p* < 0.001). A professional interpreter at the point of care was considered important by more of the migrant workers (*n* = 316) than the marital migrants (*n* = 89) (70.2 vs. 39.6%, *p* < 0.001). Although more than 90% of the surveyed migrants were enrolled in the health insurance system in Taiwan, many, especially among the migrant workers, still faced language barriers while seeking medical services.

## 1. Introduction

The population of international migrants increased from 77 million in 2000 to 244 million in 2015, and the World Health Organization (WHO) has come to view migrants’ health as an issue of gradually increasing importance [1]. Some past studies have focused on the prevalence rates of various diseases among migrants and provided evidence-based information for authorities to establish migrant healthcare policies [2,3,4,5]. In addition to such epidemiological findings, studies have also revealed various difficulties faced by migrants who require healthcare in their host countries.

In a previous study involving a health professional who attended to migrants, a language barrier existed between migrant patients and physicians but could be eliminated by trained professional interpreters [6]. A study of the Danish population found that community pharmacy staff indicated that non-Western migrant customers generally had suboptimal satisfaction when seeking assistance in healthcare-related encounters [7]. With respect to the views of migrants themselves, Turkish migrants in Germany reported being less than satisfied with and not fully understanding the information provided by physicians [8]. To lower the barriers between physicians and migrant patients, some articles have provided some practical information regarding migrants’ cultures and religions for clinicians, as well as medical education materials in different languages for migrants [9,10,11].

In Taiwan, the number of migrant workers increased from 2999 in 1991 to 676,142 in 2017 [12]. Furthermore, the number of marital migrants was around 527,000 in 2017, an increase of 12.4% in comparison to the number of marital migrants in 2012 [13]. Most of these migrant workers and marital migrants were from Southeast Asia [12,13,14]. According to the official statistics in Taiwan in 2017, 257,596 migrant workers and 29,394 marital migrants came from Indonesia, 203,613 migrant workers and 99,858 martial migrants from Vietnam, 148,475 migrant workers and 9023 marital migrants from the Philippines, and 61,543 migrant workers and 8689 marital migrants from Thailand [14,15]. Studies focusing on migrant workers in Taiwan have shown, respectively, that gastrointestinal diseases were the main reason for admission to hospitals [16], and that the frequency with which migrant workers visit clinics was related to their age, gender, insurance, and different types of jobs [17]. In addition, other studies focusing on marital migrants have revealed, respectively, that foreign brides were frustrated by language barriers and medical expenditure when visiting physicians [18], and most Vietnamese brides sought healthcare mainly because of pregnancies [19]. However, while the health-related issues of migrant workers and marital migrants in Taiwan have been generally discussed in previous research, few studies have investigated the difference between the two groups in the access to healthcare services. To fill this research gap, therefore, our study mainly analyzed the factors associated with the access of marital migrants and migrant workers to healthcare services in Taiwan.

## 2. Materials and Methods

### 2.1. Design of the Questionnaire

A structured questionnaire was developed according to previous Taiwanese studies [16,17,18,19,20,21,22,23,24,25,26,27] and was validated by three experts in this field. The questionnaire included two parts covering the basic information and personal healthcare requirements of each respondent. The questionnaire was first written in Chinese with Mandarin phonetic symbols and then translated by professional translators into Vietnamese, Indonesian, and Thai, to promote migrants who natively spoke the three languages to participate in the study because it was supposed that many of them might have problems related with access to healthcare in Taiwan where Chinese and English are the two languages mostly spoken in medical settings.

### 2.2. Setting

This study was conducted in 2018 by the Taiwan Medical Center Association and was sponsored by the Ministry of Health and Welfare. All 19 medical centers in Taiwan and 29 migrant-helping non-government associations were invited to participate in the study. Ultimately, a total of 11 medical centers and 11 migrant-helping organizations took part in the study.

### 2.3. Participants and Questionnaire Survey

The participating organizations and hospitals invited migrants from among the patients or members they served to participate in the study. In the setting of hospitals, medical personnel randomly invited participants in outpatient departments. In terms of migrant-helping organizations, staff members randomly invited participants when they served migrants. All of the participating migrants took part in the study on a voluntary basis and answered the questionnaire anonymously. First, each participant selected the version of the questionnaire in the language that he or she felt most comfortable with. It then usually took a participant 10 to 15 min to complete the questionnaire. To maintain the validity of responded questionnaires and avoid ambiguity, participants could ask assistants of the study for clarification of questions if needed. The participants received no reward or other remuneration for answering the questionnaire survey.

### 2.4. Data Collection and Extraction

A total of 1010 questionnaires were issued with 585 collected from participating hospitals and 329 from participating organizations between 1 May and 21 September 2018. After we excluded questionnaires with less than 50% of the first part (basic information) answered and those with less than 50% of the second part (personal healthcare requirements) answered, 847 effective questionnaires were extracted from the 914 questionnaires with an overall response rate of 84% (847/1010). Because migrants younger than 21 years old were less likely to have come to Taiwan for marriage or work, questionnaires filled in by respondents younger than 21 years old were excluded for analysis. In addition, questionnaires not answered by marital migrants or migrant workers were also excluded. Finally, a total of 753 surveys were analyzed. There was no statistical difference in gender distribution between the final analyzed population and the original population.

### 2.5. Data Analysis

SPSS version 23.0 (IBM, Armonk, NY, USA) was utilized for statistical analysis with a significance level of α = 0.05. To compare categorical variables between marital migrants and migrant workers in single-response questions, the Chi-square (χ^2^) test was applied as an overall test when appropriate, while Fisher’s exact test was applied whenever the χ^2^ test was not suitable. Bonferroni correction was used for post hoc tests. In multiple-response questions, the Chi-square (χ^2^) test was separately applied to the analysis of each answer.

### 2.6. Ethical Considerations

This study was approved by the Institutional Review Board of Taipei Veterans General Hospital (#2018-05-011CC).

## 3. Results

### 3.1. Characteristics of the Migrants

A total of 753 effective questionnaires were collected and analyzed (Table 1). Most of the respondents were women, while less than a quarter were men (77.1 vs. 22.9%). The percentage of marital migrants who were female is larger than that of migrant workers (95.7 vs. 66.9%, *p* < 0.001). With respect to age, 534 (70.9%) of the migrants were between 21 and 40 years old, 213 (28.3%) were between 41 and 60 years old, and six (0.8%) were older than 61 years old. The percentage of marital migrants aged between 21 and 40 years old is smaller than that of migrant workers (56.6 vs. 78.7%; *p* < 0.0167). With respect to the locations of the migrants’ dwellings or workplaces in Taiwan, 290 (39.1%) were in northern Taiwan, 87 (11.7%) were in central Taiwan, 289 (39.0%) were in southern Taiwan, and 76 (10.2%) were in eastern Taiwan. In terms of nationalities, 247 of the migrants (32.8%) were Vietnamese, 297 (39.5%) were Indonesian, 145 (19.3%) were Thai, and 63 (8.4%) had other nationalities. The percentage of marital migrants who were Vietnamese is larger than that of migrant workers (47.9 vs. 24.6%; *p* < 0.0125). For the duration of living in Taiwan, 275 of the migrants (37.2%) had lived in Taiwan less than three years, 218 (29.5%) had lived in Taiwan for three to seven years, and 246 (33.3%) had lived in Taiwan more than seven years. The percentage of marital migrants who had lived in Taiwan less than three years is smaller than that of migrant workers (10.8 vs. 51.6%; *p* < 0.0167). The proportion of marital migrants who had lived in Taiwan for three to seven years is smaller than that of migrant workers (18.1 vs. 35.7%; *p* < 0.0167). The majority of the participating migrants were covered by Taiwan’s National Health Insurance (NHI) system (*n* = 692, 93.3%), without a statistically significant difference in the rate of coverage between the marital migrants and the migrant workers. In addition to being covered by the NHI system, 344 (48.6%) of the migrants had some form of private health insurance. The percentage of marital migrants who had some form of private health insurance is larger than that of migrant workers (54.5 vs. 45.3%; *p* = 0.018). In terms of self-rated health status, 556 (74.7%) of the migrants reported good health status, 173 (23.3%) reported fair health status, and 15 (2.0%) reported poor health status. The proportion of marital migrants who had good health status is smaller than that of migrant workers (69.0 vs. 77.9%; *p* < 0.0167).

### 3.2. Chinese Language Capability

In terms of the self-evaluation of their Chinese listening ability (Table 2), 348 of the migrants (48.5%) reported having good listening ability, 247 (34.4%) reported having fair listening ability, and 123 (17.1%) reported having poor listening ability. The percentage of marital migrants who had good Chinese listening ability is larger than that of migrant workers (70.5 vs. 36.1%; *p* < 0.0167). In terms of Chinese reading ability, 147 of the migrants (22.1%) reported having good reading ability, 130 (19.6%) reported having fair reading ability, and 388 (58.3%) reported having poor reading ability. The proportion of marital migrants who had good Chinese reading ability is larger than that of migrant workers (46.5 vs. 7.9%; *p* < 0.0167). The proportion of marital migrants who had fair Chinese reading ability is larger than that of migrant workers (28.6 vs. 14.3%; *p* < 0.0167). In terms of Chinese speaking ability, 320 of the migrants (44.9%) reported having good speaking ability, 258 (36.2%) reported having fair speaking ability, and 135 (18.9%) reported having poor speaking ability. The percentage of marital migrants who had good Chinese speaking ability is larger than that of migrant workers (66.9 vs. 32.7%; *p* < 0.0167). In terms of Chinese writing ability, 105 of the migrants (15.9%) reported having good writing ability, 119 (18.0%) reported having fair writing ability, and 438 (66.1%) reported having poor writing ability. The percentage of marital migrants who had good Chinese writing ability is larger than that of migrant workers (33.9 vs. 5.5%; *p* < 0.0167). The percentage of marital migrants who had fair Chinese writing ability is larger than that of migrant workers (27.7 vs. 12.4%; *p* < 0.0167).

### 3.3. Questions Associated with Access to Healthcare Services

In terms of accessing medical services (Table 3), 304 (44.8%) of the respondents visited an outpatient department of a hospital, 278 (41.0%) visited a local western medicine clinic, 44 (6.5%) visited a pharmacy, 29 (4.3%) sought help at the emergency room of a hospital, and 23 (3.4%) visited a Chinese medicine clinic. The percentage of marital migrants who visited a local Chinese medicine clinic is larger than that of migrant workers (6.6 vs. 1.8%; *p* < 0.01). The percentage of marital migrants who visited a pharmacy is smaller than that of migrant workers (3.1 vs. 8.2%; *p* < 0.01). In terms of their experiences when seeking medical services, 247 (36.8%) of the migrants responded that they experienced language barriers. The proportion of marital migrants who experienced language barriers is smaller than that of migrant workers (17.1 vs. 48.0%; *p* < 0.001). In terms of their healthcare utilization patterns if any of the health need arises (Table 4), 385 (51.7%) answered, “I visit a physician when I have any discomfort”; 296 (39.8%) answered, “I visit a physician after my discomfort progresses”; 155 (20.8%) answered, “I buy drugs in a pharmacy when I am sick”; 109 (14.7%) answered, “I visit a physician when I need a physical checkup”; 94 (12.6%) answered, “I visit a physician when I have an injury”; 46 (8.4%) answered, “I visit a physician when I am pregnant”; and 36 (4.8%) answered, “I visit a physician when I need a medical certification”. The proportion of marital migrants who answered, “I visit a physician when I need a physical checkup” is larger than that of migrant workers (21.0 vs. 11.2%; *p* < 0.001). The proportion of marital migrants who answered, “I visit a physician when I have an injury” is larger than that of migrant workers (16.4 vs. 10.6%; *p* = 0.02). The proportion of marital migrants who answered, “I visit a physician when I am pregnant” is larger than that of migrant workers (11.6 vs. 5.9%; *p* < 0.016). In terms of someone accompanying the respondents to receive medical services, 194 (27.4%) answered none, 32 (4.5%) answered someone whose identity was not mentioned, 157 (22.2%) answered a family member, 122 (17.3%) answered an agent, 150 (21.2%) answered an employer, 52 (7.4%) answered a friend or a colleague with a foreign nationality, and 28 (4.0%) answered a Taiwanese friend or colleague. The percentage of marital migrants who visited a physician alone is larger than that of migrant workers (35.7 vs. 22.9%; *p* < 0.001). The percentage of marital migrants who visited physicians with a family member is larger than that of migrant workers (53.0 vs. 5.5%; *p* < 0.001). The percentage of marital migrants who visited physicians with an agent is smaller than that of migrant workers (0.8 vs. 26.2%; *p* < 0.001). The percentage of marital migrants who visited physicians with an employer is smaller than that of migrant workers (1.6 vs. 31.9%; *p* < 0.001). In terms of the different kinds of language assistant tools for overcoming language barriers at the point of care, 405 of the migrants (60.0%) favored having a professional interpreter available, 345 (51.1%) favored having medical sheets in their mother language, 109 (16.1%) wished for special access to medical consultations, 108 (16.0%) asked for detailed information regarding diagnoses and treatments, 105 (15.6%) wished to have indicators in different foreign languages, 103 (15.3%) wished to have access to mobile applications for real-time translations, and 94 (13.9%) wished for special clinics for migrants in their language. The proportion of marital migrants who requested professional interpretation is smaller than that of migrant workers (39.6 vs. 70.2%; *p* < 0.001. The proportion of marital migrants who requested medical sheets in their mother languages is larger than that of migrant workers (61.3 vs. 46.0%; *p* < 0.001). The proportion of marital migrants who requested detailed information about diagnoses and treatments is larger than that of migrant workers (21.3 vs. 13.3%; *p* = 0.008). The proportion of marital migrants who requested special access to medical consultations is larger than that of migrant workers (24.0 vs. 12.2%; *p* < 0.001).

## 4. Discussion

This study was designed to analyze the factors associated with the access of migrants to healthcare services and the potential differences between marital migrants and migrant workers in Taiwan. Of the migrants who participated in this study, most were women and most had come to Taiwan for marriage. These results were compatible with the phenomenon of commodified transnational marriage involving foreign brides from southeastern Asian countries marrying Taiwanese husbands that has been observed since the 1980s [28,29]. With respect to age, more of the younger migrants came to Taiwan for work than for marriage. This finding was probably associated with the labor shortages in traditional industries of Taiwan [30,31]. In terms of nationalities, most of the Vietnamese married spouses in Taiwan instead of finding jobs. The consequence was similar to that found in a previous study [32]. With respect to the duration of living in Taiwan, the migrants who had lived in Taiwan longer had mostly migrated for marriage, while those who had lived in Taiwan for shorter periods had mostly migrated for work. The reason for the former finding might be related to the attainment of permanent residency after naturalization [33], while that for the latter finding might be related to the possession of a short-term visa [34].

In terms of health insurance coverage, more than 90% of the migrants surveyed had NHI coverage when they needed medical services. The high NHI coverage rate was presumably due to the fact that migrants who have legal certificates of residence and have stayed in Taiwan for more than six months are required by law to apply for NHI coverage [35]. This result was different from that for the United States, where 14 to 44% of foreign-born adults have been reported to not be covered by any type of health insurance [36]. Furthermore, it was interesting to find that around half of the migrants surveyed in the present study had private health insurance. They might have had private insurance coverage in their home countries before coming to Taiwan for work or marriage, but the actual causes for such private insurance coverage should be further surveyed.

Compared with the marital migrants, the migrant workers surveyed in this study were more likely to report that they were healthy. This result might have been associated with the fact that migrant workers typically need to be in good health in order to do labor-intensive jobs, such as jobs in manufacturing and construction [30]. In terms of the four kinds of Chinese language abilities investigated in this study, the marital migrants reported having better abilities than the migrant workers. This was probably because many of the foreign brides had learned Chinese in order to accommodate themselves to Taiwanese society and culture after marriage [37,38].

The outpatient department of a hospital was the first choice for migrants when they were sick. In fact, the copayments for hospital-based outpatient visits are much higher than those for primary care clinic visits, and the NHI encourages patients to seek medical services at primary care clinics first [39]. However, because many large-scale hospitals provide simultaneous interpreting services through interpreters or volunteers to help international patients at the point of care, migrants often see it as easier to seek medical services at hospitals than at primary care clinics. Relatedly, earlier studies have found that language barriers between migrants and providers existed in different countries [6,9,18,40]. Marital migrants generally had less communication problems than migrant workers, probably because they needed to learn Chinese after marriage [37,38].

Unlike the healthcare systems in many other countries, the NHI system urges all employers to provide international employees with the same health insurance benefits as Taiwanese citizens once those employees have stayed longer than six months in Taiwan [16,41]. Therefore, the financial burden of healthcare coverage for migrants is relatively minimal; however, the results of this study nonetheless disclosed that a quarter of the participating migrants reported visiting physicians only when their health had deteriorated. In contrast with marital migrants, migrant workers tended to buy drugs in pharmacies when they were sick. The reason for this might be because visiting pharmacies is more convenient and time-saving for migrant workers than visiting clinics or hospitals. In addition to language barriers and communication problems, the factors related to migrants’ healthcare behaviors should be further studied.

In contrast to the migrant workers, the marital migrants surveyed in this study were inclined to visit physicians by themselves or accompanied by a family member. This result was reasonable because the marital migrants reported having better Chinese language abilities than the migrant workers and because their spouses’ families could support them. Migrant workers were mainly accompanied by their employers or agents who could act as interpreters when they asked for healthcare. However, the precision of medical interpretation by these non-professional people has not been verified, and there is no professional certification program for medical interpretation in Taiwan. As a result, a professional language interpreter was highly desired by the participating migrants, especially among the migrant workers, to help them overcome communication problems. Similar findings have also been reported in other studies, revealing that good and functional interpreters could improve the communication and the relationships between migrants and physicians [6,18,42,43].

It was difficult to find enough migrants to participate in this study on a voluntary basis without providing any reward for participating. Although we nonetheless did the best we could to recruit as many participants as possible, recruiting participants from 11 medical centers and 11 migrant-helping organizations throughout the country, there were still a few limitations in this study. First, selection bias may have existed because the survey respondents were invited from hospitals and migrant-helping organizations in Taiwan. Therefore, those who did not seek medical help or who had not accessed social welfare services would not have been recruited into this study. Second, the proportion of migrants who visited an outpatient department of a hospital might be overestimated because 64% of the responded questionnaires were gathered from participating hospitals. Third, the very few respondents who were younger than 21 were excluded from this study, and many of them were studying at junior or senior high schools. However, people in this age group people are usually healthy enough that they do not utilize medical services with great frequency. Fourth, the representativeness of the study should be gauged because the questionnaire was only translated into Vietnamese, Indonesian, and Thai, which might decrease the willingness of migrants not natively speaking the three languages to participate in the study. Related studies with a complete and detailed sampling design should be considered in the future.

## 5. Conclusions

In conclusion, although more than 90% of the migrants surveyed in this study were enrolled in the universal health insurance system in Taiwan, nearly one-third of them still had difficulty in accessing medical services due to, in part, language barriers. Migrants need to pay higher costs at hospital-based outpatient departments than at primary care clinics, so their greater use of the former may also be due to communication problems. Simultaneous interpretation services provided by professional interpreters are highly desired by migrants, especially migrant workers who do not have anyone to accompany them when seeking medical services. The results of this study are worthy of attention from healthcare and social welfare authorities insofar as they show that a large share of migrants still cannot access medical services with sufficient convenience.

## Figures and Tables

**Table 1 ijerph-16-02830-t001:** Demographic characteristics of the study population, stratified by reasons for migrating to Taiwan.

Characteristic	Marriage	Work	*p* Value
Gender (*n* = 719)			<0.001
Male ^§^	11 (4.3%)	154 (33.1%)	
Female ^§^	243 (95.7%)	311 (66.9%)	
Age (*n* = 753)			<0.001
21–40 ^§^	150 (56.6%)	384 (78.7%)	
41–60 ^§^	110 (41.5%)	103 (21.1%)	
61 and older ^§^	5 (1.9%)	1 (0.2%)	
Dwelling or workplace (*n* = 742)			<0.001
Northern Taiwan ^§^	129 (48.9%)	161 (33.7%)	
Central Taiwan ^§^	17 (6.4%)	70 (14.6%)	
Southern Taiwan	95 (36.0%)	194 (40.6%)	
Eastern Taiwan	23 (8.7%)	53 (11.1%)	
Nationality (*n* = 752)			<0.001
Vietnam ^§^	127 (47.9%)	120 (24.6%)	
Indonesia ^§^	43 (16.2%)	254 (52.2%)	
Thailand	41 (15.5%)	104 (21.4%)	
Others ^§^	54 (20.4%)	9 (1.8%)	
Duration of living in Taiwan (*n* = 739)			<0.001
Less than 3 years ^§^	28 (10.8%)	247 (51.6%)	
3 to 7 years ^§^	47 (18.1%)	171 (35.7%)	
More than 7 years ^§^	185 (71.1%)	61 (12.7%)	
National Health Insurance (*n* = 742)			0.68
Yes	243 (92.7%)	449 (93.5%)	
No	19 (7.3%)	31 (6.5%)	
Private health insurance (*n* = 708)			0.018
Yes ^§^	139 (54.5%)	205 (45.3%)	
No ^§^	116 (45.5%)	248 (54.7%)	
Self-rated health status (*n* = 744)			0.01
Good ^§^	180 (69.0%)	376 (77.9%)	
Fair ^§^	72 (27.6%)	101 (20.9%)	
Poor ^§^	9 (3.4%)	6 (1.2%)	

^§^ The column proportions were significantly different in post hoc tests with Bonferroni correction, Fisher’s exact test.

**Table 2 ijerph-16-02830-t002:** Self-reported Chinese language abilities of the study population, stratified by reasons for migrating to Taiwan.

Chinese language ability	Marriage	Work	*p* Value
Listening (*n* = 718)			<0.001
Good ^§^	182 (70.5%)	166 (36.1%)	
Fair ^§^	65 (25.2%)	182 (39.6%)	
Poor ^§^	11 (4.3%)	112 (24.3%)	
Reading (*n* = 665)			<0.001
Good ^§^	114 (46.5%)	33 (7.9%)	
Fair ^§^	70 (28.6%)	60 (14.3%)	
Poor ^§^	61 (24.9%)	327 (77.8%)	
Speaking (*n* = 713)			<0.001
Good ^§^	170 (66.9%)	150 (32.7%)	
Fair ^§^	69 (27.2%)	189 (41.2%)	
Poor ^§^	15 (5.9%)	120 (26.1%)	
Writing (*n* = 662)			<0.001
Good ^§^	82 (33.9%)	23 (5.5%)	
Fair ^§^	67 (27.7%)	52 (12.4%)	
Poor ^§^	93 (38.4%)	345 (82.1%)	

^§^ The column proportions were significantly different in post hoc tests with Bonferroni correction.

**Table 3 ijerph-16-02830-t003:** Single-response questions associated with access of the study population to healthcare, stratified by reasons for migrating to Taiwan.

Question	Marriage	Work	*p* Value
Accessibility to medical services (*n* = 678)			0.002
An outpatient department in a hospital	102 (44.9%)	202 (44.8%)	
An emergency room in a hospital	8 (3.5%)	21 (4.6%)	
A local western medicine clinic	95 (41.9%)	183 (40.6%)	
A local Chinese medicine clinic ^§^	15 (6.6%)	8 (1.8%)	
A pharmacy ^§^	7 (3.1%)	37 (8.2%)	
Language barriers existed when I visited a physician (*n* = 672)			<0.001
Yes ^§^	42 (17.1%)	205 (48.0%)	
No ^§^	203 (82.9%)	222 (52.0%)	

^§^ The column proportions were significantly different in post hoc tests with Bonferroni correction.

**Table 4 ijerph-16-02830-t004:** Multiple-response questions associated with access of the study population to healthcare, stratified by reasons for migrating to Taiwan.

Question	Marriage	Work	*p* Value
Behavior of healthcare utilization if any of the health need arises (*n* = 744)			
I visit a physician when I have any discomfort	142 (54.2%)	243 (50.4%)	0.32
I visit a physician after my discomfort progresses	106 (40.5%)	190 (39.4%)	0.78
I buy drugs in a pharmacy when I am sick	47 (17.9%)	108 (22.4%)	0.15
I visit a physician when I need a physical checkup	55 (21.0%)	54 (11.2%)	<0.001
I visit a physician when I have an injury	43 (16.4%)	51 (10.6%)	0.02
I visit a physician when I am pregnant (*n* = 548)	28 (11.6%)	18 (5.9%)	0.016
I visit a physician when I need a medical certification	10 (3.8%)	26 (5.4%)	0.34
A person who can accompany me to a clinic visit (*n* = 707)			
None	89 (35.7%)	105 (22.9%)	<0.001
Someone whose identity was not mentioned	11 (4.4%)	21 (4.6%)	0.92
A family member	132 (53.0%)	25 (5.5%)	<0.001
An agent	2 (0.8%)	120 (26.2%)	<0.001
An employer	4 (1.6%)	146 (31.9%)	<0.001
A friend or a colleague with a foreign nationality	12 (4.8%)	40 (8.7%)	0.06
A Taiwanese friend or colleague	11 (4.4%)	17 (3.7%)	0.65
Different kinds of language assistant tools which can help me utilize healthcare (*n* = 675)			
An available professional interpreter	89 (39.6%)	316 (70.2%)	<0.001
Medical sheets in my mother language	138 (61.3%)	207 (46.0%)	<0.001
Detailed information about diagnoses and treatments	48 (21.3%)	60 (13.3%)	0.008
Special access to medical consultations	54 (24.0%)	55 (12.2%)	<0.001
Hospital indicators in different foreign languages	43 (19.1%)	62 (13.8%)	0.07
Mobile applications for real-time translations	41 (18.2%)	62 (13.8%)	0.13
Special clinics for migrants	36 (16.0%)	58 (12.9%)	0.27
Others	18 (8.0%)	11 (2.4%)	0.001

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
