# Peer review of "Factors Associated with Access of Marital Migrants and Migrant Workers to Healthcare in Taiwan: A Questionnaire Survey with Quantitative Analysis"

_ijerph, 2019, doi:10.3390/ijerph16162830_

Round 1
Reviewer 1 Report
The manuscript describes a questionnaire survey that explored the characteristics of marital and working migrants in relation to healthcare utilization and potential language barriers. Considering the large number of migrants who come to Taiwan for marriage and work and their contribution to Taiwan’s economy and society, ensuring their welfare through timely and equitable access to health care is of great importance. This study will provide some valuable insight for health care provision for these migrants.
Major comments
1. Statistical analysis: Pages 6-7, Table 3: It appears that multiple responses were allowed for the questions “Behavior of healthcare utilization” and “Different kinds of language assistant tools which can help me utilize healthcare”. For these two questions, would it be more appropriate to calculate percentages for individual items using the total number of respondents as the denominator rather than using total number of responses as the denominator? Comparisons using chi squared tests or Fisher’s exact tests can be made item by item rather than carrying out an overall test across all items, which might not be valid as the responses are not mutually exclusive for these two questions?
2. Wording of questionnaire: Would it be possible for the authors to include either the original questionnaire in Chinese language or an English translation of the questionnaire as an online appendix? It is difficult for readers to interpret the findings without seeing the original questions. For example, for the question “Behavior of healthcare utilization”, were the respondents asked about their general intention if any of the health condition/need arises, or were they asked about the reason(s) for the specific visit to the hospital/migrant-helping organization where they were given the questionnaire?
3. Influence of location of sampling: the questionnaire was administered in medical centres and migrant-helping organizations, and therefore the characteristics of the survey participants are likely to be influenced by these. Could the author indicate how many/what proportions of the questionnaires were obtained from medical centres and migrant-helping organizations respectively? Would the finding that a high proportion of respondents chose hospital for their health care partly be attributed to the sampling method? If so, this should be acknowledged in the Discussion section.
4. Representativeness of study sample: would it be possible to provide some information from official statistics in the Introduction or Discussion section concerning the number of marital and working migrants from each of the main neighbouring countries (i.e. Vietnam, Indonesia, Thailand, the Philippines) so that readers can gauge the representativeness of the study sample? A good proportion of foreign labours come from the Philippines. Would their inclusion in this survey be affected by the available languages of the questionnaire? If so, this should also be acknowledged in the Discussion section.
Minor comments
5. Title: I wonder if “…A questionnaire survey with quantitative analysis” would make the title more informative.
6. Page 2, lines 86-87: please state how many migrant-helping non-government associations were invited.
7. I have made various suggestions in relation to the use of terms/English language for the authors to consider:
Page 3, lines 114-115: perhaps it is more accurate to say “Most of the ‘respondents’ were women”?
Page 3, lines 119 &128; page 5, lines 162 & 165: … a smaller percentage
Page 3, lines 129 & 137; page 6, line 172: -- a smaller proportion
Author Response
Response to Reviewer 1 Comments
Dear Reviewer 1,
We sincerely wrote the responses to your comments in this letter.
Point 1:
Statistical analysis: Pages 6-7, Table 3: It appears that multiple responses were allowed for the questions “Behavior of healthcare utilization” and “Different kinds of language assistant tools which can help me utilize healthcare”. For these two questions, would it be more appropriate to calculate percentages for individual items using the total number of respondents as the denominator rather than using total number of responses as the denominator? Comparisons using chi squared tests or Fisher’s exact tests can be made item by item rather than carrying out an overall test across all items, which might not be valid as the responses are not mutually exclusive for these two questions?
Response 1:
Thanks for the reviewer’s helpful suggestion. In fact, multiple responses were allowed for the questions “Behavior of healthcare utilization”, “A person who can accompany me to a clinic visit”, and “Different kinds of language assistant tools which can help me utilize healthcare”. Based on the suggestion, we divided the original Table 3 into a new Table 3 and Table 4. Single-response questions were displayed in the new Table 3 and multiple-response questions were in the new Table 4 as follows:
Table 3. Single-response questions associated with access of the study population to healthcare, stratified by reasons for migrating to Taiwan | |||
Marriage | Work | p value | |
Accessibility to medical services (n = 678) | 0.002 | ||
An outpatient department in a hospital | 102 (44.9%) | 202 (44.8%) | |
An emergency room in a hospital | 8 (3.5%) | 21 (4.6%) | |
A local western medicine clinic | 95 (41.9%) | 183 (40.6%) | |
A local Chinese medicine clinic § | 15 (6.6%) | 8 (1.8%) | |
A pharmacy § | 7 (3.1%) | 37 (8.2%) | |
Language barriers existed when I visited a physician (n = 672) | <0.001 | ||
Yes § | 42 (17.1%) | 205 (48.0%) | |
No § | 203 (82.9%) | 222 (52.0%) | |
§ The column proportions were significantly different in post hoc tests with Bonferroni correction |
Table 4. Multiple-response questions associated with access of the study population to healthcare, stratified by reasons for migrating to Taiwan | |||
Marriage | Work | p value | |
Behavior of healthcare utilization if any of the health need arises (n = 744) | |||
I visit a physician when I have any discomfort | 142 (54.2%) | 243 (50.4%) | 0.32 |
I visit a physician after my discomfort progresses | 106 (40.5%) | 190 (39.4%) | 0.78 |
I buy drugs in a pharmacy when I am sick | 47 (17.9%) | 108 (22.4%) | 0.15 |
I visit a physician when I need a physical checkup | 55 (21.0%) | 54 (11.2%) | <0.001 |
I visit a physician when I have an injury | 43 (16.4%) | 51 (10.6%) | 0.02 |
I visit a physician when I am pregnant | 29 (11.1%) | 19 (3.9%) | <0.001 |
I visit a physician when I need a medical certification | 10 (3.8%) | 26 (5.4%) | 0.34 |
A person who can accompany me to a clinic visit (n = 707) | |||
None | 89 (35.7%) | 105 (22.9%) | <0.001 |
Someone whose identity was not mentioned | 11 (4.4%) | 21 (4.6%) | 0.92 |
A family member | 132 (53.0%) | 25 (5.5%) | <0.001 |
An agent | 2 (0.8%) | 120 (26.2%) | <0.001 |
An employer | 4 (1.6%) | 146 (31.9%) | <0.001 |
A friend or a colleague with a foreign nationality | 12 (4.8%) | 40 (8.7%) | 0.06 |
A Taiwanese friend or colleague | 11 (4.4%) | 17 (3.7%) | 0.65 |
Different kinds of language assistant tools which can help me utilize healthcare (n = 675) | |||
An available professional interpreter | 89 (39.6%) | 316 (70.2%) | <0.001 |
Medical sheets in my mother language | 138 (61.3%) | 207 (46.0%) | <0.001 |
Detailed information about diagnoses and treatments | 48 (21.3%) | 60 (13.3%) | 0.008 |
Special access to medical consultations | 54 (24.0%) | 55 (12.2%) | <0.001 |
Hospital indicators in different foreign languages | 43 (19.1%) | 62 (13.8%) | 0.07 |
Mobile applications for real-time translations | 41 (18.2%) | 62 (13.8%) | 0.13 |
Special clinics for migrants | 36 (16.0%) | 58 (12.9%) | 0.27 |
Others | 18 (8.0%) | 11 (2.4%) | 0.001 |
In the new Table 4, the total number of respondents was used as the denominator to calculate percentages for individual items. Furthermore, chi squared tests were made item by item. To illustrate methods for analysis of single-response and multiple-response questions to readers, sentences in the Method section 2.5. were revised as follows:
To compare categorical variables between marital migrants and migrant workers in single-response questions, the Chi-square (χ2) test was applied as an overall test when appropriate, while Fisher’s exact test was applied whenever the χ2 test was not suitable. Bonferroni correction was used for post hoc tests. In multiple-response questions, the Chi-square (χ2) test was separately applied to analysis of each answer.
In the Result section 3.3., the sentences related to the new Table 4 were revised as follows:
In terms of their healthcare utilization patterns (table 4), 385 (51.7%) answered, “I visit a physician when I have any discomfort”; 296 (39.8%) answered, “I visit a physician after my discomfort progresses”; 155 (20.8%) answered, “I buy drugs in a pharmacy when I am sick”; 109 (14.7%) answered, “I visit a physician when I need a physical checkup”; 94 (12.6%) answered, “I visit a physician when I have an injury”; 48 (6.5%) answered, “I visit a physician when I am pregnant”; and 36 (4.8%) answered, “I visit a physician when I need a medical certification”. The proportion of marital migrants who answered, “I visit a physician when I need a physical checkup” is larger than that of migrant workers (21.0% vs. 11.2%; p<0.001). The proportion of marital migrants who answered, “I visit a physician when I have an injury” is larger than that of migrant workers (16.4% vs. 10.6%; p=0.02). The proportion of marital migrants who answered, “I visit a physician when I am pregnant” is larger than that of migrant workers (11.1% vs. 3.9%; p<0.001). In terms of someone accompanying the respondents to receive medical services, 194 (27.4%) answered none, 32 (4.5%) answered someone whose identity was not mentioned, 157 (22.2%) answered a family member, 122 (17.3%) answered an agent, 150 (21.2%) answered an employer, 52 (7.4%) answered a friend or a colleague with a foreign nationality, and 28 (4.0%) answered a Taiwanese friend or colleague. The percentage of marital migrants who visited a physician alone is larger than that of migrant workers (35.7% vs. 22.9%; p<0.001). The percentage of marital migrants who visited physicians with a family member is larger than that of migrant workers (53.0% vs. 5.5%; p<0.001). The percentage of marital migrants who visited physicians with an agent is smaller than that of migrant workers (0.8% vs. 26.2%; p<0.001). The percentage of marital migrants who visited physicians with an employer is smaller than that of migrant workers (1.6% vs. 31.9%; p<0.001). In terms of the different kinds of language assistant tools for overcoming language barriers at the point of care, 405 of the migrants (60.0%) favored having a professional interpreter available, 345 (51.1%) favored having medical sheets in their mother language, 109 (16.1%) wished for special access to medical consultations, 108 (16.0%) asked for detailed information regarding diagnoses and treatments, 105 (15.6%) wished to have indicators in different foreign languages, 103 (15.3%) wished to have access to mobile applications for real-time translations, and 94 (13.9%) wished for special clinics for migrants in their language. The proportion of marital migrants who requested professional interpretation is smaller than that of migrant workers (39.6% vs. 70.2%; p<0.001). The proportion of marital migrants who requested medical sheets in their mother languages is larger than that of migrant workers (61.3% vs. 46.0%; p<0.001). The proportion of marital migrants who requested detailed information about diagnoses and treatments is larger than that of migrant workers (21.3% vs. 13.3%; p=0.008). The proportion of marital migrants who requested special access to medical consultations is larger than that of migrant workers (24.0% vs. 12.2%; p<0.001).
Point 2:
Wording of questionnaire: Would it be possible for the authors to include either the original questionnaire in Chinese language or an English translation of the questionnaire as an online appendix? It is difficult for readers to interpret the findings without seeing the original questions. For example, for the question “Behavior of healthcare utilization”, were the respondents asked about their general intention if any of the health condition/need arises, or were they asked about the reason(s) for the specific visit to the hospital/migrant-helping organization where they were given the questionnaire?
Response 2:
Thanks for the reviewer’s constructive suggestion. To avoid ambiguity and to make readers truly understand the findings in the study, the original questionnaire in Chinese language would be included as an online appendix. Furthermore, the questionnaire would be sent to the assistant editor because only one file could be uploaded on the web page of “Author's Reply to the Review Report”. The question “Behavior of healthcare utilization” were that the respondents asked about their general intention if any of the health condition/need arises.
To clarify the question “Behavior of healthcare utilization” for readers, the clause “In terms of their healthcare utilization patterns if any of the health need arises” was used in the Result section 3.3. and the clause “Behavior of healthcare utilization if any of the health need arises” was used in the new Table 4.
Point 3:
Influence of location of sampling: the questionnaire was administered in medical centres and migrant-helping organizations, and therefore the characteristics of the survey participants are likely to be influenced by these. Could the author indicate how many/what proportions of the questionnaires were obtained from medical centres and migrant-helping organizations respectively? Would the finding that a high proportion of respondents chose hospital for their health care partly be attributed to the sampling method? If so, this should be acknowledged in the Discussion section.
Response 3:
Thanks for the reviewer’s constructive suggestion. The number of questionnaires from medical centers and that from migrant-helping organizations were written in the Method section 2.4. in the revised manuscript as follows:
A total of 1,010 questionnaires were issued with 585 collected from participating hospitals and 329 from participating organizations between May 1st and September 21st, 2018.
According to a higher proportion of questionnaires from hospitals, which might be attributed to the finding that a high proportion of respondents chose hospital for their health care, we wrote the sentences in the limitation section of the Discussion as follows:
Second, the proportion of migrants who visited an outpatient department of a hospital might be overestimated because 64% of the responded questionnaires were gathered from participating hospitals.
Point 4:
Representativeness of study sample: would it be possible to provide some information from official statistics in the Introduction or Discussion section concerning the number of marital and working migrants from each of the main neighbouring countries (i.e. Vietnam, Indonesia, Thailand, the Philippines) so that readers can gauge the representativeness of the study sample? A good proportion of foreign labours come from the Philippines. Would their inclusion in this survey be affected by the available languages of the questionnaire? If so, this should also be acknowledged in the Discussion section.
Response 4:
Thanks for the reviewer’s detailed suggestion. We wrote official statistics concerning the number of marital and working migrants from each of the main neighbouring countries in the Introduction section in the revised manuscript as follows:
According to the official statistics in Taiwan in 2017, 257,596 migrant workers and 29,394 marital migrants came from Indonesia, 203,613 migrant workers and 99,858 martial migrants from Vietnam, 148,475 migrant workers and 9,023 marital migrants from the Philippines, and 61,543 migrant workers and 8,689 marital migrants from Thailand [14, 15].
In the Method section 2.1., we explained the reason why the questionnaire was translated into Vietnamese, Indonesian, and Thai in the revised manuscript as follows:
The questionnaire was first written in Chinese with Mandarin phonetic symbols and then translated by professional translators into Vietnamese, Indonesian, and Thai to promote migrants who natively spoke the three languages to participate in the study because it was supposed that many of them might have problems related with access to healthcare in Taiwan where Chinese and English are the two languages mostly spoken in medical settings.
As what the reviewer thought, the survey could be affected by the available languages of the questionnaire. Therefore, we wrote the sentences in the limitation section of the Discussion as follows:
Fourth, the representativeness of the study should be gauged because the questionnaire was only translated into Vietnamese, Indonesian, and Thai, which might decrease the willingness of migrants not natively speaking the three languages to participate in the study. Related studies with a complete and detailed sampling design should be considered in the future.
Point 5:
Title: I wonder if “…A questionnaire survey with quantitative analysis” would make the title more informative.
Response 5:
Thanks for the reviewer’s detailed suggestion. The title was revised as “Factors Associated with Access of Marital Migrants and Migrant Workers to Healthcare in Taiwan: A Questionnaire Survey with Quantitative Analysis.”
Point 6:
Page 2, lines 86-87: please state how many migrant-helping non-government associations were invited.
Response 6:
Thanks for the reviewer’s detailed suggestion. The sentences were written in the Method section 2.2. in the revised manuscript as follows:
All 19 medical centers in Taiwan and 29 migrant-helping non-government associations were invited to participate in the study.
Point 7:
I have made various suggestions in relation to the use of terms/English language for the authors to consider:
Page 3, lines 114-115: perhaps it is more accurate to say “Most of the ‘respondents’ were women”?
Page 3, lines 119 &128; page 5, lines 162 & 165: … a smaller percentage
Page 3, lines 129 & 137; page 6, line 172: -- a smaller proportion
Response 7:
Thanks for the reviewer’s detailed suggestion. The sentence, “Most of the respondents were women, while less than a quarter were men (77.1% vs. 22.9%)” was written in the Result section 3.1. in the revised manuscript.
“Less” was replaced with “smaller” for the comparative in the Result section. In addition, according to the other reviewer suggestion, sentences structuring as “A less percentage of the marital migrants than the migrant workers were aged between 21 and 40 years old” were hard to understand. Therefore, the structure of these sentences was revised as “The percentage of marital migrants aged between 21 and 40 years old is smaller than that of migrant workers”. In the new sentence structure, “less” was replaced with “smaller” for the comparative.
Thank you for your attention and constructive comments to our study.
Reviewer 2 Report
I appreciate your efforts to contribute to the important topic of migration and health. However, I do have some comments.
Introduction:
-Sentences are too long.
- This section could be improved by examining a little more on what is known about this topic in literature.
Methods:
-Did the participants require help in filling the questionnaires?
-Who invited the migrants to participate in the study?
Results:
-The structuring of these sentences makes them difficult to understand. I suggest that you edit all sentences using this format.
‘A less percentage of the marital migrants than the migrant workers were aged between 21 and 40 years old’.
‘A less percentage of the marital migrants than the migrant workers had lived in Taiwan less than 3 years (10.8% vs. 51.6%; p<0.0167)’.
‘A less proportion of the marital migrants than the migrant workers had lived in Taiwan for 3 to 7 years (18.1% vs. 35.7%; p<0.0167).’
-And many such sentences in the text.
Discussion:
-Are there unmarried spouses?
-This study, though not a qualitative study, is focused on human experience. What validity criteria were used to ensure rigour of data collected and analysed?
-What does this manuscript add to literature?
Author Response
Response to Reviewer 2 Comments
Dear Reviewer 2,
We sincerely wrote the responses to your comments in this letter.
Point 1:
Introduction:
-Sentences are too long.
-This section could be improved by examining a little more on what is known about this topic in literature.
Response 1:
Thanks for the reviewer’s constructive suggestion. To shorten sentences and to add other literature related to the topic, we deleted some original sentences and clauses less associated with healthcare utilization, access to healthcare, and migrants in the revised manuscript, such as “Cancer was the leading cause of death among Asian immigrants to America, followed by heart disease and cerebrovascular disease [3]”, “On the other hand, essential hypertension and visual disturbance were the top two chronic diseases among Iraqi refugees in Jordan [4]”, “midwives in Australia reported that refugees to the country faced language and cultural barriers and had difficulty in receiving healthcare [6]”, “A study of Malaysia, meanwhile, found that the main barriers faced by refugees and asylum-seekers in utilizing medical resources were related to a lack of health knowledge, a lack of awareness of the right to healthcare, illegal statuses, and low incomes [7]”, “while sub-Saharan refugees in Australia faced language barriers, poor financial support, limited health-related information, and insufficient access to healthcare when they needed help [10]”, “male workers in the construction and manufacturing industries had a high risk of occupational injury [18]”, and “the occurrence of hereditary diseases among foreign brides was worthy of investigation [20], and that the behavior of health promotion was correlated with background factors [21].” Other sentences and clauses associated with healthcare utilization, access to healthcare, and migrants were added in the revised manuscript, such as “In a previous study involving a health professional who have attended to migrants, a language barrier existed between migrant patients and physicians but could be eliminated by trained professional interpreters [6], and “In addition, other studies focusing on marital migrants have revealed, respectively, that foreign brides were frustrated by language barriers and medical expenditure when visiting physicians [18], and most Vietnamese brides sought for healthcare mainly because of pregnancies [19]”.
After revision, the Introduction section was written as follows:
The population of international migrants increased from 77 million in 2000 to 244 million in 2015, and the World Health Organization (WHO) has come to view migrants’ health as an issue of gradually increasing importance [1]. Some past studies have focused on the prevalence rates of various diseases among migrants and provided evidence-based information for authorities to establish migrant healthcare policies [2-5]. In addition to such epidemiological findings, studies have also revealed various difficulties faced by migrants who require healthcare in their host countries.
In a previous study involving a health professional who have attended to migrants, a language barrier existed between migrant patients and physicians but could be eliminated by trained professional interpreters [6]. A study of Danish found that community pharmacy staff indicated that non-Western migrant customers generally had suboptimal satisfaction when seeking assistance in healthcare-related encounters [7]. With respect to the views of migrants themselves, Turkish migrants in Germany reported being less than satisfied with and not fully understanding the information provided by physicians [8].To lower the barriers between physicians and migrant patients, some articles have provided some practical information regarding migrants’ cultures and religions for clinicians, as well as medical education materials in different languages for migrants [9-11].
In Taiwan, the number of migrant workers increased from 2,999 in 1991 to 676,142 in 2017 [12]. Furthermore, the number of marital migrants was around 527,000 in 2017, an increase of 12.4% in comparison to the number of marital migrants in 2012 [13]. Most of these migrant workers and marital migrants were from Southeast Asia [12-14]. According to the official statistics in Taiwan in 2017, 257,596 migrant workers and 29,394 marital migrants came from Indonesia, 203,613 migrant workers and 99,858 martial migrants from Vietnam, 148,475 migrant workers and 9,023 marital migrants from the Philippines, and 61,543 migrant workers and 8,689 marital migrants from Thailand [14, 15]. Studies focusing on migrant workers in Taiwan have shown, respectively, that gastrointestinal diseases were their main reason for admission to hospitals [16], and that the frequency with which migrant workers visit clinics was related to their age, gender, insurance, and different types of jobs [17]. In addition, other studies focusing on marital migrants have revealed, respectively, that foreign brides were frustrated by language barriers and medical expenditure when visiting physicians [18], and most Vietnamese brides sought for healthcare mainly because of pregnancies [19]. However, while the health-related issues of migrant workers and marital migrants in Taiwan have been generally discussed in previous research, few studies have investigated the difference between the two groups in the access to healthcare services. To fill this research gap, therefore, our study mainly analyzed the factors associated with the access of marital migrants and migrant workers to healthcare services in Taiwan.
Point 2:
Methods:
-Did the participants require help in filling the questionnaires?
Response 2:
Thanks for the reviewer’s detailed suggestion. If some participants required help in filling the questionnaires, assistants of the study would give assistance. The process was written in the Method section 2.3. in the revised manuscript as follows:
To maintain the validity of responded questionnaires and avoid ambiguity, participants could ask assistants of the study for clarification of questions if needed.
Point 3:
Methods:
-Who invited the migrants to participate in the study?
Response 3:
Thanks for the reviewer’s detailed suggestion. People who invited the migrants were written in the Method section 2.3. in the revised manuscript as follows:
In the setting of hospitals, medical personnel randomly invited participants in outpatient departments. In terms of migrant-helping organizations, staff members randomly invited participants when they served migrants.
Point 4:
Results:
-The structuring of these sentences makes them difficult to understand. I suggest that you edit all sentences using this format.
‘A less percentage of the marital migrants than the migrant workers were aged between 21 and 40 years old’.
‘A less percentage of the marital migrants than the migrant workers had lived in Taiwan less than 3 years (10.8% vs. 51.6%; p<0.0167)’.
‘A less proportion of the marital migrants than the migrant workers had lived in Taiwan for 3 to 7 years (18.1% vs. 35.7%; p<0.0167).’
-And many such sentences in the text.
Response 4:
Thanks for the reviewer’s constructive suggestion. To make the sentences less difficult to understand, they were edited in the revised manuscript according to a native English speaker’s suggestion.
“A less percentage of the marital migrants than the migrant workers were aged between 21 and 40 years old.” was rewritten as “The percentage of marital migrants aged between 21 and 40 years old is smaller than that of migrant workers.”
“A less percentage of the marital migrants than the migrant workers had lived in Taiwan less than 3 years (10.8% vs. 51.6%; p<0.0167)” was rewritten as “The percentage of marital migrants who had lived in Taiwan less than 3 years is smaller than that of migrant workers (10.8% vs. 51.6%; p<0.0167).”
“A less proportion of the marital migrants than the migrant workers had lived in Taiwan for 3 to 7 years (18.1% vs. 35.7%; p<0.0167)” was rewritten as “The percentage of marital migrants who had lived in Taiwan for 3 to 7 years is smaller than that of migrant workers (18.1% vs. 35.7%; p<0.0167).”
Other sentences with the same structure as “A less percentage of the marital migrants than the migrant workers were aged between 21 and 40 years old” were all rewritten in the Result section in the revised manuscript.
Point 5:
Discussion:
-Are there unmarried spouses?
Response 5:
Thanks for the reviewer’s detailed question. To be honest, we did not know the information about unmarried spouses because different types of spouses were not surveyed in the questionnaires.
Point 6:
Discussion:
-This study, though not a qualitative study, is focused on human experience. What validity criteria were used to ensure rigour of data collected and analysed?
Response 6:
Thanks for the reviewer’s detailed suggestion. The method used to ensure rigour of data collected and analysed was written in the Method in the revised manuscript as follows:
To maintain the validity of responded questionnaires and avoid ambiguity, participants could ask assistants of the study for clarification of questions if needed (in the Method section 2.3.).
A total of 1,010 questionnaires were issued with 585 collected from participating hospitals and 329 from participating organizations between May 1st and September 21st, 2018. After we excluded questionnaires with less than 50% of the first part (basic information) answered and those with less than 50% of the second part (personal healthcare requirements) answered, 847 effective questionnaires were extracted from the 914 questionnaires with an overall response rate of 84% (847/1,010) (in the Method section 2.4.).
Point 7:
Discussion:
-What does this manuscript add to literature?
Response 7:
Thanks for the reviewer’s detailed question. In the end of the Introduction section, we mentioned that our study would add “the difference between marital migrants and migrant workers in the access to healthcare services” to current literature. The main finding added to current literature was showed in the Discussion section in the manuscript as follows:
Migrant workers were mainly accompanied by their employers or agents who could act as interpreters when they asked for healthcare. However, the precision of medical interpretation by these non-professional people has not been verified, and there is no professional certification program for medical interpretation in Taiwan. As a result, a professional language interpreter was highly desired by the participating migrants, especially among the migrant workers, to help them overcome communication problems.
Thank you for your attention and constructive comments to our study.
Round 2
Reviewer 1 Report
Thank you for the opportunity to re-review the paper. The authors have satisfactorily addressed my previous comments, and the revised manuscript is much improved. I have only one remaining comment, which the authors might wish to consider:
As the question 'I visit a physician when I am pregnant' was applicable only to female, the denominators might need to be adjusted in the calculation of percentages for this specific question?
Author Response
Thanks for the reviewer's detailed suggestion. The replay was presented in the attachment.
